# Analyses of the *MYBL1* Gene in Triple Negative Breast Cancer: Evidence of Regulation of the *VCPIP1* Gene and Identification of a Specific Exon Overexpressed in Tumor Cell Lines

**DOI:** 10.3390/ijms26010279

**Published:** 2024-12-31

**Authors:** Chidinma Nganya, Sahia Bryant, Ayah Alnakhalah, Taylor Allen-Boswell, Sierra Cunningham, Samuel Kanu, Ashton Williams, Deshai Philio, Kathy Dang, Emmanuel Butler, Audrey Player

**Affiliations:** Department of Biology, Texas Southern University, Houston, TX 77004, USA; c.nganya2651@student.tsu.edu (C.N.); s.bryant6199@student.tsu.edu (S.B.); a.alnakhalah0034@student.tsu.edu (A.A.); t.allenboswell9040@student.tsu.edu (T.A.-B.); sierracu@usc.edu (S.C.); s.kanu4742@student.tsu.edu (S.K.); a.williams9002@student.tsu.edu (A.W.); d.philio3155@student.tsu.edu (D.P.); dang1kathy@gmail.com (K.D.); e.butler6406@student.tsu.edu (E.B.)

**Keywords:** triple negative breast cancer (TNBC), V-Myb Avian Myeloblastosis Viral Oncogene Homolog-Like 1 (MYBL1), Valosin-Containing Protein P97/P47 Complex-Interacting Protein 1 (VCPIP1), tumor-associated exons

## Abstract

Previous data show that the knockdown of the *MYBL1* gene in the MDA-MB-231 cell line leads to the downregulation of *VCPIP1* gene expression. In addition, *MYBL1* and *VCPIP1* genes are co-expressed and dysregulated in some of the same triple negative breast cancer patient samples. We propose that the co-expression of the two genes is attributed to the MYBL1 transcription factor regulation of the *VCPIP1* gene. We identify the MYBL1 transcription factor binding site upstream of the VCPIP1 start site and show that the MYBL1 protein can bind to the sequence identified in the VCPIP1 promoter region. Combined with the results from the knockdown study, these data support the ability of *MYBL1* to regulate the *VCPIP1* gene. The *VCPIP1* gene functions as a deubiquitinating enzyme involved in DNA repair, protein positioning, and the assembly of the Golgi apparatus during mitotic signaling. The transcriptional regulation of VCPIP1 by the *MYBL1* gene could implicate MYBL1 in these processes, which might contribute to tumor processes in TNBC. Although both genes are involved in cell cycle regulatory mechanisms, converging signaling mechanisms have not been identified. In a separate study, we performed sequence alignment of the MYBL1 transcript variants and identified an exon unique to the canonical variant. Probes that specifically target the unique *MYBL1* exon show that the exon is overexpressed in tumor cell lines compared to non-tumor breast cells. We are classifying this unique MYBL1 exon as a tumor-associated exon.

## 1. Introduction

In the United States of America alone, in the year 2024, its estimated that 2,001,140 new patients will be diagnosed with cancer. Of these cases, 310,720 will be invasive breast cancers, with an estimated 10–15% of these patients diagnosed with triple negative breast cancer (TNBC) [1]. While effective targeted therapies exist for receptor-positive breast cancers, fewer targeted therapeutic options exist for patients with TNBC. More recently, immunotherapy has been approved and shows promise in the treatment of some TNBCs [2]. Even with effective therapy, it is still critically important to understand the underlying molecular events that drive TNBC processes so that additional biomarkers can be identified.

The *MYBL1* gene belongs to the MYB family of transcriptional activators, which includes *c-MYB* and *MYBL2* genes [3]. The genes were initially identified based on their homology to the v-myb oncogene carried by avian leukemia viruses and the E26 leukemia virus. As a family, the *c-MYB*, *MYBL1* and *MYBL2* genes are involved in cell proliferation, differentiation, apoptosis, and tumor transformation processes [4,5]. Specifically related to *MYBL1*, it is one of several genes associated with the DREAM complex, which functions to repress genes involved in cell cycle regulation [6,7,8]. DREAM stands for ‘dimerization partner, RB-like, E2F, and multi-vulva class B (MuvB)’. The complex includes the *E2F4*, *E2F5*, *LIN9*, *LIN37*, *LIN52*, *LIN54*, *MYBL1*, *MYBL2*, *RBL1*, *RBL2*, *RBBP4*, *TFDP1*, and *TFDP2* genes, of which *MYBL1* is a member. Collectively the genes are master regulators of the cell cycle at stages G1/S and G2/M, where the RBL1 and RBL2 are key regulators in the protein complex. During the repressive and activating phases of the cell cycle, various members of the complex function by binding or dissociating with each other, leading to the regulation of the cell cycle. Recent data show that both MYBL1 and MYBL2 can bind MuvB (which includes the LIN9, LIN37, LIN52, LIN54, and RBBP4 complexes), leading to the downstream recruitment of the other cell cycle regulators, the *CDK1*, *CCNB1*, and *FOXM1* genes [7,9]. These data demonstrate the importance of MYBL1 in cell cycle regulation, a key process related to the pathogenesis of tumors. Related to cancers, the *MYBL1* gene is over-expressed in low-grade gliomas [10], dysregulated in breast adenoid cystic carcinomas (a rare triple negative breast cancer (TNBC)) and salivary gland carcinomas where gene-fusion mutations with *ACTN1* and *NFIB* genes are identified [11]. The *MYBL1* gene is also over-expressed in clear cell renal carcinoma and considered an immunotherapeutic biomarker for these cancers [12]. Studies of clear cell renal carcinoma show that *MYBL1* expression correlations with the immune scores, increasing Tregs, M2 macrophages, neutrophils, B cells, monocytes, and CD8+ T cells. The over-expression of *MYBL1* also correlates with the overexpression of the key immune checkpoint genes *PD-1*, *CTLA4*, *PD-L1*, and *PD-L2* in clear cell renal cell carcinoma patients. The authors suggest that *MYBL1* can ultimately remodel the tumor micro-environment, but it is unclear how *MYBL1* might enhance the cellular malignant behaviors of clear cell renal carcinomas.

In an earlier study, we identified the *MYBL1* gene overexpressed in a subpopulation of patient samples and TNBC cell lines [13]. Because the gene is a strong transcriptional activator and involved in many of the events key to tumor progression, we knocked down MYBL1 in MDA-MB-231 cells and identified *MYBL1* gene partners either directly or indirectly affected by the process [14]. In addition to MYBL1, which is located at the chromosomal 8q13.1 locus, we identified a substantial number of genes affected by the knockdown process which were also localized to the 8q loci [15], including the MYC transcription factor and the *VCPIP1* gene.

Several studies demonstrate co-expression between *MYBL1* and *MYC* genes. The *MYC* and *MYBL1* genes function synergistically to affect the cell’s entry into the S phase of the cell cycle [16,17,18]. And *MYC* is a downstream target of *MYBL1* related to apoptotic events relevant to the pathology of human B-cell neoplasia [17,19]. In contrast, other than the identification of a *MYBL1-VCPIP1* gene fusion event in salivary gland tumors [20], a possible relationship between *MYBL1* and *VCPIP1* genes has not been established. The *VCPIP1* gene is involved in the assembly and disassembly of the Golgi apparatus during the cell cycle events and deubiquitylation events related to DNA repair [21]. The *VCPIP1* gene is particularly interesting to our research because it is (a) downregulated when we silence MYBL1 in MDA-MB-231 cells, (b) it is located at the 8q13.1 chromosomal locus, like MYBL1, (c) it is dysregulated in many of the same TNBC patient samples as MYBL1, (d) bioinformatic analyses indicate MYBL1 transcription factor protein binding to the *VCPIP1* gene, and (e) both genes are involved in cell cycle signaling mechanisms. In the current study, we consider the possibility that the aforementioned relationships between MYBL1 and VCPIP1 are based on the ability of the MYBL1 transcription factor to bind directly to the VCPIP1 promoter region, thus affecting the gene’s expression. We identify the MYBL1 binding site in the *VCPIP1* promoter and find that the MYBL1 protein can indeed bind to this site.

As of November 2024, three known curated Reference MYBL1 transcript variants have been deposited at the National Center for Biotechnology Information (NCBI; [22]) website, along with additional predicted Reference Sequences. We performed comparative sequence alignment of the individual MYBL1 exons and identified a unique exon 15 associated with the canonical NM_001080416.4 transcript variant 1 compared to the other known Reference sequences. The exon 15 corresponds to a region immediately down-stream of the Negative Regulatory region of MYB proteins, at the carboxy terminal of the proteins, in a region susceptible to post-translational modification [23]. Because of the unique differences between the transcript variants, we considered the possibility that certain variants were associated with non-tumor or TNBC. So, the second phase of the current study involved analyses of the expression of the MYBL1 exon 15 in TNBC. We find that compared to the non-tumor breast cell line, the exon 15 nucleotides and corresponding regions in the protein isoform are overexpressed in the TNBC cells.

## 2. Results

### 2.1. Co-Expression of MYBL1 and VCPIP1 Genes and Experimental Analyses of MYBL1 Transcription Factor Binding to VCPIP1 Promoter

When the *MYBL1* gene is silenced in MDA-MB-231 cells, VCPIP1 transcript expression is also downregulated, suggesting that MYBL1 is either directly or indirectly responsible. As validation, we continue to examine both genes in cell lines (Figure 1) and patient samples [13,15] and find co-expression of the genes in TNBC samples. In 11 independent PCR experiments, the difference between the mean densitometer values for VCPIP1 expression in MCF10A control versus TNBC was 0.28 compared to 0.78, respectively, with a *p*-value of 0.02. And the difference between the mean densitometer values for MYBL1 expression in MCF10A control versus TNBC was 0.25 versus 0.61, respectively, with a *p*-value of 0.03.

TNBC patient samples were extracted from the Breast Invasive Carcinoma TCGA 2015 [24] and METABRIC [25] datasets. The TCGA dataset was chosen because it contained various types of mutational, proteome, and transcriptome analyses of known genes in clinical patient samples. The box plots in Figure 2A demonstrate the types of mutational and transcriptome alterations and the patterns of alterations observed in MYBL1 compared to *VCPIP1* genes in the TNBC patient samples. Similar patterns of mutations are observed for both genes in the patient samples. A similar pattern of shallow deletions, copy number gains and amplifications are observed for *MYBL1* and *VCPIP1* genes. An Oncoprint plot of these data are summarized in Figure 2B. The same 10 patients displayed amplifications, and ~64% of the patients showed alterations in both *MYBL1* and *VCPIP1* genes (Figure 2B). Only gene amplifications were identified in the METABRIC patient dataset (Figure 2C). Fifty-one of 320 patients (i.e., 16%) show amplifications in both *MYBL1* and *VCPIP1* genes in the METABRIC dataset. The number of patients with alterations is low; still, these data demonstrate complete concordance between alterations in *MYBL1* and *VCPIP1* genes. Many of the patients did not demonstrate alterations in either gene, as designated by ‘no alterations’ in the diagram.

Our previous data comparing the expression of *MYBL1* and *VCPIP1* genes support a possible functional relationship between the genes in certain TNBC cells. In addition, because bioinformatic analyses suggest the possible regulation of VCPIP1 by the MYBL1 transcription factor, we decided to experimentally examine the possibility that MYBL1 binds directly to the *VCPIP1* promoter region, explaining our previous gene expression results. *MYBL1* is a strong transcriptional activator capable of regulating and altering the transcription of a number of genes [5,26]. Other than the fact that *MYBL1* and *VCPIP1* genes are associated with cell cycle signaling events and a MYBL1-VCPIP1 fusion product has been observed in salivary gland tumors, no experimental data support a relationship between the two genes.

Online bioinformatic resources were searched to determine if computer-predicted interactions exist between the MYBL1 protein and the *VCPIP1* promoter regions. A search of the GeneHancer^TM^ program [27] listed the *VCPIP1* gene as a possible target for the MYBL1 transcription factor (https://www.genecards.org/cgi-bin/carddisp.pl?gene=VCPIP1&keywords=vcpip1 (accessed on 12 July 2024)). Using resources available on the NCBI website, we identified the *VCPIP1* promoter region upstream of the gene’s start site on chromosome 8 (Figure 3). The start site for the VCPIP1 protein is designated by the ATG nucleotide sequence. Because transcription factors (like MYBL1) bind to the promoter region, the *VCPIP1* promoter region was retrieved and validated by sequence alignment analyses. The BLAST program [22] was utilized to validate the *VCPIP1* promoter sequence given in Figure 3. The sequence given in Figure 3 aligns 100% with the chromosome 8 locus corresponding to the *VCPIP1* promoter and 25% with the actual *VCPIP1* gene (Figure 4). The entire *VCPIP1* gene was not analyzed via BLAST; only 25% of the *VCPIP1* gene was analyzed using BLAST. The MYBL1 transcription factor binding sequence was retrieved from the JASPAR database ([28]; Figure 5A). And the Alggen Promo program [29] was used to identify two putative *MYBL1* binding sites in regions upstream of the *VCPIP1* start site (Figure 5B).

Before a transcription factor can regulate the expression of a particular gene, the protein must first bind to the promoter region of the targeted gene. The Electrophoretic Mobility Shift Assay (EMSA) is performed to show that the MYBL1 transcription factor can bind to the promoter region of the *VCPIP1* gene. The purified MYBL1 protein only bound to one of the putative *VCPIP1* sequences (Figure 6), the sequence given in Figure 5B. To validate the performance of the EMSA assay, as a control, the Epstein–Barr virus protein was shown to bind to its specific DNA probe sequence (lane 1). Lane 2 contains the biotin end-labeled VCPIP1 probe given in Figure 5B. Without MYBL1, the biotin-labeled VCPIP1 probe migrates as a low-molecular-weight sequence (lane 2), but following the incubation with and binding of the unlabeled MYBL1 recombinant protein, the mobility of the VCPIP1 fragment shows a shift in mobility and higher-molecular-weight migration of the probe. These data show that the MYBL1 protein can bind to the *VCPIP1* promoter region. An increase in the concentration of a ‘cold unlabeled VCPIP1 probe binding to the unlabeled MYBL1 protein’ was utilized as the EMSA control to demonstrate a lack in the ‘shift in the mobility of the VCPIP1 probe’ when high concentrations of the unlabeled probe compete with the binding MYBL1 protein. Experimental results of the EMSA assay show that the purified MYBL1 recombinant protein binds to the sequence ~940 nucleotides upstream of the VCPIP1 start site. These results support the MYBL1 protein’s ability to bind to the promoter region of the *VCPIP1* gene.

### 2.2. Identification of a Unique Exon Associated with MYBL1 Transcript Variants and Protein Isoforms

Splice variants are functional alternatives of a gene caused by spliceosomes composed of small nuclear ribonucleoproteins (snRNPs) [30]. Based on data deposited at the NCBI and comparative sequence alignments, there are 17 MYBL1 known and predicted Reference sequence transcript variants, 10 of which are unique. Of the 10 Reference sequences, 3 are known, curated MYBL1 transcript variants (designated NM) and 7 are predicted transcript variants (designated XM). The known transcript variants include NM_001080416.4 (NM1), NM_001144755.3 (NM2), and NM_001294282.2 (NM3) (Figure 7). Following the comparative sequence alignment of the three known variants, we find that NM1, also referred to as the canonical sequence, is the longest variant. NM1 also contains a unique exon 15 lacking in the other 2 transcript variants. The current study focuses on the further characterization of the MYBL1 exon 15 in the known Reference Sequence. Figure 7 demonstrates the intron/exon distribution of the three known MYBL1 Reference transcript variants, NM1, NM2, and NM3 show the location of the unique exon 15. Compared to NM2 and NM3, only NM1 contains the exon 15 sequence; all other nucleotide regions of the variants are nearly perfectly aligned (Figure 8A). NM3, however, differs from NM1 and NM2 by three nucleotides at the beginning of exon 9, at position 1250 of the nucleotide sequence, which codes for the glutamine amino acid in the protein sequence. The exon 15 region contains 180 unique nucleotides. The arrows in the figure highlight the positions of the left and right PCR primers generated to determine the expression of exon 15 in non-tumor compared to TNBC cell lines via the PCR procedure (addressed below). Figure 8B shows the position for the amino acids corresponding to the exon 15 nucleotides. The oval region outlines the unique amino acids coded from MYBL1 exon 15, also addressed below. PhosphoSite-PLus^TM^ [31] was utilized to examine the possible relevance of the amino acid region corresponding to the exon 15 region. Data show that the region is susceptible to post-translational modification via phosphorylation, ubiquitination, and acetylation (Figure 9).

Data show that individual isoforms for a particular gene can have different functions in the same type of sample, emphasizing the importance of characterizing the isoforms related to particular genes [33]. We screened several commercially available MYBL1 antibodies and found that antibodies that recognized the region corresponding to exon 15 demonstrated the consistent differential detection of TNBC samples. At this stage of the project, we do not know how exon 15 contributes to the tumor process, only that exon 15 is over-represented in the TNBC compared to the non-tumor breast cells. For any particular gene, generally, the transcript variants are functionally related. The variants might have opposing functions, but generally, there is some relationship. For the *MYBL1* gene, we suspect that the function of variants without or with exon 15 differ based on transcriptional activation because the *MYBL1* gene is a transcription factor.

*MYBL1* PCR gene primers were designed such that they specifically captured the exon 15 transcript variant region (Figure 8A). The corresponding amplicon is 212 nucleotides. Transcript analyses show that exon 15 is differentially overexpressed in the MDA-MB-436 and MDA-MB-231 TNBC cell lines compared to the non-tumor estrogen receptor-negative MCF10A cells, suggesting over-expression of the exon in TNBC compared to the non-tumor breast cell line (Figure 10A). Protein expression levels were also examined in the breast cell line samples. Two different MYBL1 antibodies were utilized to compare protein expression levels of the exon 15 region in the breast cell lines. Both MYBL1 antibodies recognized amino acids corresponding to the exon 15 target sequence which contains 60 amino acids. However, one was a commercial antibody which included the exon 15 region in Figure 8B, with an additional 20 or so amino acids on either side of the target region; this final immunogen totaled 110 amino acids, which are given in the Methods section. Additionally, the other MYBL1 antibody was a custom antibody which only included the 60 amino acids corresponding to the unique exon 15 sequences designated in Figure 8B. Similar to transcript analyses, protein analyses show differential overexpression in the TNBC cell lines compared to the non-tumor breast cell line (Figure 10B). These data confirm that the region corresponding to exon 15 is over-expressed in TNBC cells compared to non-tumor cells. The VCPIP1 protein levels were also included in these analyses to show over-expression of the protein in TNBC compared to non-tumor breast cells.

We show that the MYBL1 protein can bind to the *VCPIP1* promoter, but we have yet to demonstrate the possible signaling events that connect the genes. Summarized in the STRING^TM^ analyses [34], Interactome data demonstrate protein:protein interactions between VCPIP1 and MAF1, and the Reactome database shows evidence of interactions between the MAF1 and RBBP4 genes (Figure 11). RBBP4 is part of the DREAM complex [6,8], thus establishing a connection with MYBL1. The STRING^TM^ PPI (protein:protein interaction) enrichment *p*-value is 0.00123. PPI represents all manners of associations, including co-expressions, direct (physical) and indirect (functional), and predicted and known interactions, including data collected from curated databases. At this point, all of these interactions have not been experimentally validated, so the STRING^TM^ data represent predicted associations between the *VCPIP1* and *MYBL1* genes.

## 3. Discussion

The *MYBL1* gene belongs to a family of v-Myb like protooncogenes. While the MYB family of genes displays some of the same functions, they also display distinctly different transcriptional regulatory functions [5]. Data integrated across various organisms suggest that the MYBL1 transcription factor exerts its function via binding to DNA to activate transcription through RNA polymerase II (Alliance of Genome Resources [30]). Early data show that MYBL1 is a master regulator of male meiosis [36], regulating genes essential for germline integrity. With co-operation with c-MYC, the *MYBL1* gene can rescue human B-cell neoplasia from apoptosis, aiding in tumor progression [37]. A substantial number of publications validate a role of MYBL1 in different phases of cell cycle regulation, interacting with genes related to the DREAM complex and individual cell cycle-related genes [6,7,8,38]. We identified a number of genes on the 8q loci far from and near the MYBL1 locus (i.e., 8q13.1), affected by MYBL1 knockdown in MDA-MB-231 cells; many of these genes are related to cell cycle-related functions [15].

The *VCPIP1* gene is one of many genes at the 8q loci affected by the knockdown of MYBL1 in MDA-MB-231 cells. The *VCPIP1* gene is located at the 8q13.1 locus and is a deubiquitinating enzyme involved in DNA repair and membrane fusion events related to the reassembly of the Golgi apparatus and the endoplasmic reticulum after mitosis [21]. Other than the fact that both *MYBL1* and *VCPIP1* genes are at the 8q13.1 locus and function in cell cycle signaling pathways, there are almost no data suggesting a relationship between the two genes. However, Bubola et al. [20] identified a MYBL1-VCPIP1 fusion product following the genomic characterization of salivary gland tumors. The authors suggest that this and other fusion events contribute to the pathogenesis of salivary gland cancers. The results from our laboratory show the experimental validation of co-expression between the MYBL1 and *VCPIP1* genes in TNBC. And we identify the MYBL1 binding sequence in the *VCPIP1* promoter region and demonstrate the ability of the MYBL1 protein to bind to this sequence, all steps necessary for the regulation of gene expression.

Other than data presented in this experiment and our previous experiments, we have not been able to confidently establish a relationship between MYBL1 and VCPIP1 signaling processes. Investigators show that *VCPIP1* interacts with *VCP* and other genes associated with the reassembly of the Golgi apparatus and the endoplasmic reticulum during mitosis [39,40]. We examined the possibility that MYBL1 might be associated with this signaling pathway, but the experimental results were inconsistent. Further analyses of our MYBL1 knockdown data did, however, show downregulation of the MAF1 gene. The MAF1 gene is relevant because several different interactome analyses demonstrate VCPIP1 and MAF1 protein:protein interactions [35] (https://www.ncbi.nlm.nih.gov/gene/80124 (accessed on 25 June 2024)). The MAF1 gene plays a key role in cell fate determination and is a negative regulator of RNA polymerase III [41]. We are considering the possibility that *MYBL1* and *VCPIP1* are linked via the *MAF1* gene in *TNBCs*. If *MYBL1* and *VCPIP1* are linked via *MAF1*, it might be via (a) the *MYC* gene, which is known to be associated with MYBL1 [37] and/or (b) the RBBP4 histone-binding protein, which is one of several genes associated with MYBL1 in the DREAM complex [6]. STRING^TM^ analyses of the five genes *MYBL1*, *VCPIP1*, *MAF1*, *MYC*, and *RBBP4* display a PPI enrichment *p*-value of 0.00123, indicating a significant chance the genes are known or predicted to be functionally related. A relationship with the MYC gene is supported by (a) GeneHancer^TM^ and Qiagen promoter analyses which show that c-MYC is a top transcription factor regulating MAF1 and (b) our data, which show both MYC and MAF1 downregulated with MYBL1 silencing. It could be that the *MAF1* gene is indirectly affected by MYBL1 in these processes. Reactome analyses show an interaction between RBBP4 and MAF1, and RBBP4 and MYBL1 are part of the DREAM complex. The other four genes are affected by our knockdown procedure, and at the 8q loci, RBBP4 is not. We are in the process of examining these and other possible signaling connections between *MYBL1* and *VCPIP1* gene expression. Still, based on our current and previous results, we can conclude that MYBL1 binds directly to the *VCPIP1* promoter, leading to the regulation of its gene expression in some TNBCs.

As part of the current study, we also examined MYBL1 transcript variants and their resulting protein isoforms. Transcript variants are the result of splicing events by snRNP protein–RNA complexes. There are many predicted MYBL1 Reference Sequence variants deposited at NCBI, but according to the Mammalian Gene Collection (MGC; [42]), two known Reference Sequence MYBL1 transcript variants exist, where NM2 and NM3 are considered to be the same sequence. Because we are interested in understanding how MYBL1 might contribute to the phenotype and signaling processes related to TNBC, we began by characterizing the individual transcript variants and protein isoforms. Our data show that MYBL1 transcript variants (and the corresponding protein isoforms) differ only by a unique exon 15 present in the canonical NM1 sequence. This makes NM1 180 nucleotides longer than the other variant(s). The unique sequence is downstream from the MYBL1 Negative Regulatory Region, at the carboxy terminal of the gene, in a region susceptible to regulation by the post-translational modification by ubiquitylation, acetylation, and phosphorylation. When probes that uniquely capture this region are utilized to determine transcript and protein levels, the over-expression of MYBL1 exon 15 is observed in the TNBC cell line samples. Marginal transcript expression levels are observed in non-tumor breast cells. The results from these experiments suggest that the transcript variant expressing the exon 15 is preferentially associated with tumors. A functional relationship has yet to be established in our case, but there are documented instances where unique exons exist in tumor versus non-tumor cells, where the unique exons are regarded as tumor-associated.

Different isoforms of the same gene are thought to be an evolutionary strategy for promoting molecular diversity and/or exist where one isoform, via competitive binding, can modulate the function of the other in vivo. Wang and Kirkness [43] suggest that in mammals, variations in the number of exons lead to lineage-specific evolution. Nearly all genes exist as transcript variations and subsequent protein isoforms, and data show that the variations of the particular gene do not always have the same function. Gene variations are generally related to splicing events that lead to differences in the exon pattern of the gene, leading to disease processes, including associations with tumors [30,33]. Under these conditions, the exons are described as tumor-associated exons. Researchers understand the slicing mechanisms that lead to variations related to exon differences, but not the underlying events that trigger the processes. Once the triggering event has occurred, in the case of cancers, it could be that at the early stage of tumor progression, there were higher levels of the non-tumor variants compared to variants expressing tumor-associated exons. And with later stages of progression, there is a shift such that there are higher levels of the variants expressing the tumor-associated exons. Regardless, the function of the tumor-associated exon clearly relies upon the function of the gene.

Guttery et al. ref [44] identified a truncated form of the Tenascin-C transcript expressed in normal breast tissue and a larger molecular weight transcript variant of the gene over-expressed in the tumor tissues. Similar to what we observed for the *MYBL1* gene, exons 14 and 16 of the Tenascin-C gene are considered tumor-associated regions of the gene that affect invasive properties of the gene in breast cell lines. Because a similar observation was observed in other cancers, investigators suggest that exons 14 and 16 of Tenascin-C be considered as markers of prognosis and invasion in various cancers [45,46,47]. In the case of the Bcl-X gene, two isoforms exist, which demonstrate divergent properties. There is a longer Bcl-xL and a shorter Bcl-xS [48] isoform of the gene. The larger Bcl-xL isoform is related to survival and the smaller isoform Bcl-xS activates apoptosis, proving that different forms of the same gene can have opposing functions. As an explanation, Gasparski et al. [49] show that different variants and protein isoforms are processed differently in the nucleus and cytoplasm, based on their 3’ untranslated regions and coding sequences, which lead to different biological functions.

The data presented here document the presence of exon 15 over-expression in some TNBCs, but we are aware that TNBCs are extremely heterogenous cancers. As a continuum of the current study, we will examine whether the extra exon 15 is overexpressed in other TNBC cells and TNBC patient samples. Second, we are attempting to validate the association of the shorter variants with non-tumor TNBCs. Lastly, we are not entirely clear about the relationship between the expression of the longer variant and the cell’s proliferative process. We are currently addressing these concerns.

## 4. Materials and Methods

### 4.1. Maintenance of Cell Lines

The MCF10A, MDA-MD-436, and MDA-MB-231 cell lines cell lines utilized for this study were obtained from the American Type Culture Collection (ATCC^®^) (Manassas, VA, USA). The MCF10A cell line is a non-tumor estrogen receptor negative breast cell line, while the MDA-MB-436 and MDA-MB-231 cells are characterized as mesenchymal stem-like TNBC cell lines. The cell lines were maintained according to ATCC recommendations. The cells were cultured in T75 tissue culture dishes, fed biweekly with Dulbecco’s Modified Eagle Minimum Essential Media (DMEM; catalog # 11965092, Thermo-Fisher Scientific, Norristown, PA, USA) supplemented with 1% penicillin (catalog # 15140122, Thermo-Fisher Scientific, Norristown, PA, USA) and 10% serum (catalog # A5670701, Thermo-Fisher Scientific, Norristown, PA, USA) and incubated in a 37 °C incubator with 5% CO_2_. The cells were allowed to reach approximately 80–90% confluence, typsinized using a 0.25% trypsin-EDTA solution (catalog # 25300054, Thermo-Fisher Scientific, Norristown, PA, USA) and then sub-cultured or used for experiments.

### 4.2. TNBC Patient Sample Dataset Analyzed for MYBL1 and VCPIP1 Gene Expression

Two clinical patient datasets were extracted from cBioPortal.org [50] and utilized for gene expression analyses. The Breast Invasive Carcinoma TCGA 2015 dataset [24] was extracted from cBioPortal.og and examined using their online analyses platform. Eighty-three TNBC samples were extracted from the dataset, and the genes were analyzed for various types of mutational, proteome, and transcriptome alterations. 

Three hundred and twenty TNBC patient samples were also extracted from The Breast Cancer METABRIC dataset [25] also obtainable at cBioPortal.org (https://www.cbioportal.org; TCGA and the METABRIC datasets were last accessed on 11 June 2024). 

### 4.3. Ribonucleic Acid (RNA) Isolation and Analyses

Total RNA was extracted from the cell lines using 1 mL of TRIzol^TM^ solution, following the manufacturer’s guidelines (catalog # 15596026, Thermo-Fisher Scientific, Norristown, PA, USA). The purified RNA was resuspended in clean water and analyzed by spectrophotometer analyses. Preparations with an A260/280 ration of 1.8–2.0 were utilized for downstream analyses. The total RNA profiles were examined via 3-(N-morpholino) propane sulfonic acid (MOPs; catalog 351-059-101, Quality Biologicals, Gaithersburg, MD, USA) agarose gel electrophoresis.

### 4.4. Generating Complementary DNA (cDNA)

The Bio-Rad iScript^TM^ copy DNA (cDNA) kit (catalog #1708890, Bio-Rad, Hercules, CA, USA) was used to generate cDNA from purified total RNA. The procedure was performed according to the manufacturer’s suggestion. The resulting cDNA was stored at –20 °C prior to use for polymerase chain reaction (PCR) analyses, or at –80 °C for long-term storage.

### 4.5. Generation and Validation of the PCR Gene Primer Sets

The nucleotide sequences corresponding to the GAPDH (ID 2597) control gene and *MYBL1* (ID 9603) and *VCPIP1* genes (ID 80124) were retrieved from the NCBI database. The Primer3^TM^ program was used to generate PCR primer sequences specifically designed to detect the target genes ([51]; version 0.4.0; https://bioinfo.ut.ee/primer3-0.4.0/ (accessed on 2 May 2024)). The original MYBL1 primer set aligned with the 3’ untranslated region of the nucleotide based on a sequence present on the DNA microarray platform. Primer sets generated using Primer3^TM^ and their corresponding amplicons were further evaluated using the NCBI Basic Local Alignment Search Tool [22] and the In-silico PCR analysis program of the University of California Santa Cruz Genome Browser ([52]; https://genome.ucsc.edu/ (accessed on 2 May 2024)). The primer sets were synthesized by IDTDNA.com (Coralville, IA, USA). The primer sequences and the corresponding amplicon sizes for the target genes are given in Table 1.

### 4.6. PCR Procedure

Approximately 10 nanograms (ng) of cDNA were combined with 10 μL of a 2× Amplitaq Gold™ (catalog #4311806, Life Technologies, Waltham, MA, USA) mixture, 2 μL of the combined forward and reverse gene primers (i.e., about 1 µM), and water to a final volume of 20 μL, which was used for each PCR reaction. The PCR conditions included an initial denaturation at 95 °C for 5 min, followed by 30 cycles of 95 °C for 30 s, 56 °C for 90 s, and 70 °C for 90 s, with a final extension at 70 °C for 5 min. The PCR products were separated on a 1.0% agarose gel.

### 4.7. Western Blotting Procedure and Reagents

The Western blot procedure was conducted as described in a previous document [53]. The antibodies and dilution concentrations used in this study are listed below: Actin anti-rabbit antibody (catalog # SAB14002280; Sigma Aldrich/Millipore, St. Louis, MO, USA) was used at a 1:1000 dilution. The MYBL1 anti-mouse antibody (catalog #SAB14002280; Sigma Aldrich/Millipore, St. Louis, MO, USA) was applied at a 1:1000 dilution; the immunogen sequence was KSLVLDNWEKEESGTQLLTEDISDMQSENRFTTSLLMIPLLEIHDNRCNLIPEKQDINSTNKTYTLTKKKPNPNTSKVVKLEKNLQSNCEWETVVYGKTEDQLIMTEQAR. This is a commercial MYBL1 antibody containing exon 15-associated amino acids plus 20 or so additional amino acids on either side of the region. Amino acids corresponding to the exon 15 region are underlined.

The MYBL1 exon 15 specific antibody was generated as a custom polyclonal antibody by the Sino Biological company (Wayne, PA, USA) and utilized at a dilution of 1 ug/uL. The custom antibody for the *MYBL1* gene was based on the amino acid sequence corresponding to exon 15 only. The following amino acid sequence corresponds to the exon 15 nucleotide sequence: ENRFTTSLLMIPLLEIHDNRCNLIPEKQDINSTNKTYTLT-KKKPNPNTSKVVKLEKNLQS. Note that this immunogen is the same as the amino acids underlined for the commercial MYBL1 antibody. The sequence was made available to Sino Biotechnologies company. The company generated the amino acid sequence, injected it into rabbits, bled the rabbits after 6 months, purified the IgG antibody, and shipped the antibody protein to our university. Based on the immunogen, only variants with amino acids corresponding to the exon 15 region are detected. only protein isoforms containing the immunogen sequence should bind the antibody. Full-length MYBL1 with exon 15 is ~85,885 Daltons. The western blot was performed as validation, and we observed an 85,000 Dalton protein band corresponding to the MYBL1 isoform for the protein containing amino acids corresponding to the exon 15 region. The molecular weight for the protein was determined by comparison to the Precision Plus Protein™ Dual Color Standard molecular weight standard (catalog # 1610374; Bio-Rad, Hercules, CA, USA).

The VCPIP1 anti-mouse antibody (catalog # sc-515281) was obtained from Santa Cruz Biotechnology (Santa Cruz, CA, USA) and utilized at a 1:200 dilution. Secondary HRP-conjugated anti-mouse (catalog #HAF007) and anti-rabbit (catalog #HAF008) antibodies were purchased from R&D Systems (Minneapolis, MN, USA) and utilized at a 1:1000 dilution. Western blotting nitro-cellulose filters were developed using the Clarity Western ECL substrate (catalog # 1705061, Bio-Rad, Hercules, CA, USA) and imaged and analyzed using the LI-COR digital imaging system (LI-COR Biotechnology, Lincoln, NE, USA).

### 4.8. Identification and Validation of the MYBL1 Transcription Factor Binding Site in the VCPIP1 Promoter

The MYBL1 transcription factor binding sequence was identified using the 10th release of the JASPAR online database ([28]; accessed on 2 April 2023; https://jaspar.elixir.no/). DNA regions upstream of the VCPIP1 protein start site were identified using the NCBI gene analyses tool and validated using the Basic Local Alignment Search Tool (BLAST; [22]; GRCH38.14 Primary Assembly). The purported MYBL1 binding region in the *VCPIP1* promoter was identified using the Alggen Promo online promoter analyses program ([29]; version 3; accessed on 12 June 2023). The GeneHancer^TM^ online tool ([27]; Assembly GRCh38/hg38) available at GeneCards.org was used to identify genes that were predicted or experimentally determined to be MYBL1 targets. The double stranded MYBL1 transcription factor binding sequence (i.e., the VCPIP1 probe) was made available to Creative-Proteomics for use in the Electrophoretic Mobility Shift Assay (EMSA; [54]).

### 4.9. EMSA Reagents and Proczedure

The *VCPIP1* promoter probe sequence utilized for the EMSA was identified at Texas Southern University and made available to Creative-Proteomics. All other supplies for the procedure and the EMSA procedure were performed by the Creative-Proteomics company (Shirley, NY, USA). In summary, the MYBL1 recombinant protein was generated, expanded, purified and validated by Creative-Proteomics. Creative-Proteomics prepared the biotin-labeled and cold (unlabeled) VCPIP1 probes, which were annealed to the complementary oligonucleotides in the presence of an annealing buffer. Cold VCPIP1 probes were used to compete with binding to the MYBL1 protein. The MYBL1 protein was incubated with VCPIP1-biotin-labeled probes at room temperature for up to 15 min. A 6% native polyacrylamide gel was prepared and pre-run to ensure uniform separation. The binding reactions were loaded onto the gel and run at 95 Volts. After electrophoresis, the gel was transferred to a nylon membrane using an electrophoretic transfer system, and the transferred DNA was crosslinked to the membrane using UV light. The positive Epstein bar-body probe, plus the Epstein probe with the positive Epstein protein, were utilized as controls.

### 4.10. Sequence Alignment and Other Data Analyses

Nucleotide and protein sequences were retrieved from the NCBI database. The MultAlin^TM^ program ([32]; http://multalin.toulouse.inra.fr/multalin/ (accessed on 12 July 2024)) was used to perform the sequence alignment comparisons between MYBL1 transcript variants or protein isoforms. The MultAlin program was last modified 3 February 2000. The MYBL1 Negative Regulatory region and regions susceptible to post-translational modifications were defined using the PhosphoSite-PLus^TM^ online database resource ([31]; version.6.7.5; https://www.phosphosite.org/homeAction.action (accessed on 12 July 2024)). STRING^TM^ (Search Tool for Recurring Instances of Neighbouring Genes) ([34]; version 12.0; https://string-db.org/ (accessed on 12 July 2024)) was utilized to identify predicted and known relationships between genes. The University of California Genome Browser (GRCH38.14 Primary Assembly; [52]) and GeneCards.org ([55]; version 5.22; https://www.genecards.org/ (accessed on 12 July 2024)) were also utilized as scientific data analyses resources.

## 5. Conclusions

In a previous study, we identified *MYBL1* and *VCPIP1* genes co-expression in some TNBC cell lines and patient samples. As validation, we show that the co-expression of genes can be due to the ability of the MYBL1 transcriptional factor to bind to possibly directly regulate VCPIP. We identify the MYBL1 transcription factor binding site in the *VCPIP1* promoter and show that the MYBL1 protein can bind to this site. In a separate but related analysis of MYBL1 expression, we identify a unique exon 15 in the canonical sequence. Data show that this exon is over-represented in TNBC compared to non-tumor breast cells. We are aware that TNBCs are extremely heterogenous cancers. As a continuum of the current study, we will examine whether the extra exon 15 is overexpressed in other TNBC cells and TNBC patient samples. Second, we are attempting to validate the association of the shorter variants with non-tumor TNBCs. Lastly, we are not entirely clear about the relationship between the expression of the longer variant and the cell’s proliferative process. We are currently addressing these concerns.

## Figures and Tables

**Figure 1 ijms-26-00279-f001:**
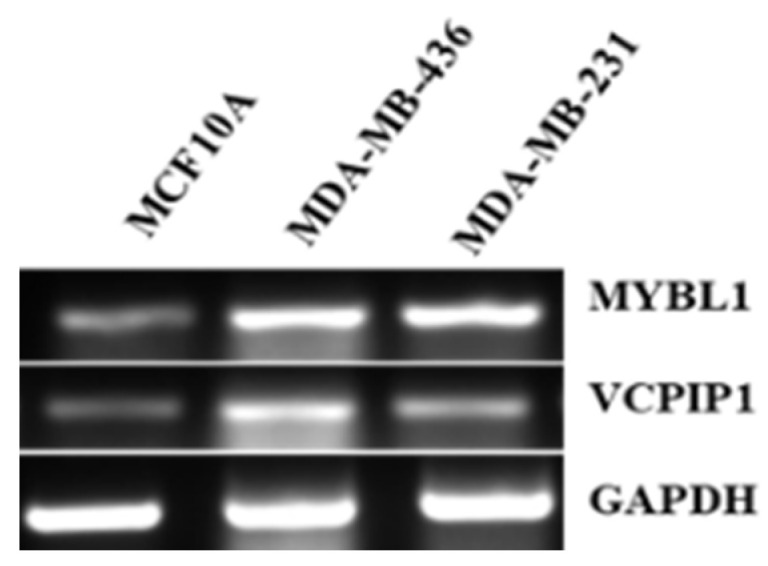
PCR analyses of MYBL1 and VCPIP1 transcript levels in non-tumor (MCF10A) vs. MDA-MB-436 and MDA-MB-231 (TNBC) cells. The original MYBL1 primer set was utilized for this analysis.

**Figure 2 ijms-26-00279-f002:**
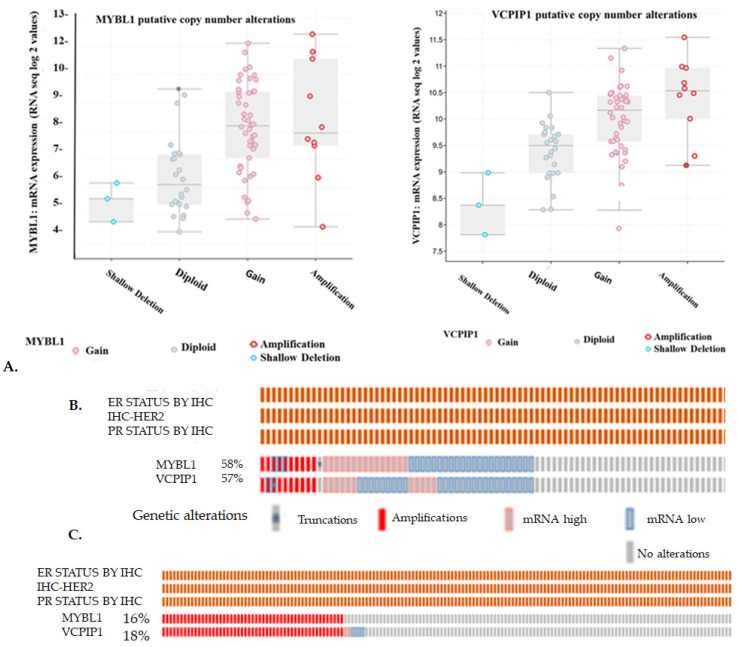
Analyses of *MYBL1* and *VCPIP1* expression in TNBC patient samples extracted from cBioPortal. (**A**) Box plot analyses of 83 TCGA patients examined for various types of alterations in *MYBL1* and *VCPIP1* genes [24]. (**B**) Oncoprint analyses of the 83 TCGA TNBC patient samples analyzed for alterations in *MYBL1* and *VCPIP1* genes. (**C**) A total of 320 TNBC samples from the METABRIC [25] dataset were extracted as TNBC and examined for *MYBL1* and *VCPIP1* gene alterations. The entire list of samples demonstrating ‘no detection’ are not displayed (i.e., gray boxes). A total of 172 of the 320 samples are displayed. Each bar represents a different patient. The legend designates the type of alterations identified in the *MYBL1* and *VCPIP1* genes. The percentage values represent the number of patient samples analyzed for MYBL1 or VCPIP1 alterations in the particular dataset.

**Figure 3 ijms-26-00279-f003:**
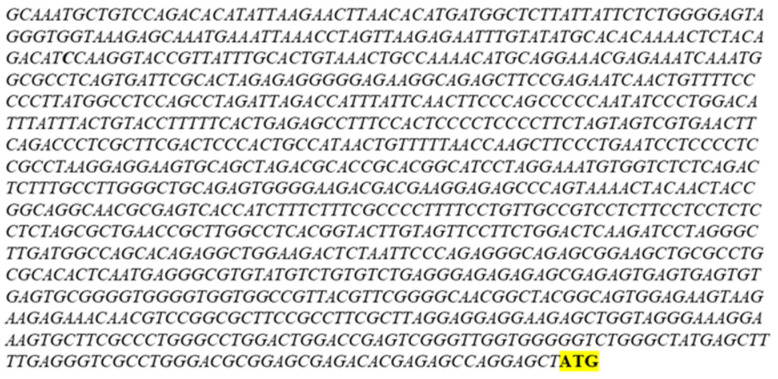
Analyses of the *VCPIP1* promoter and 5’ untranslated genomic region. The *VCPIP1* promoter and the 5’ untranslated region of the gene are shown in italics. The protein ATG start site is highlighted in yellow. The nucleotide sequence was retrieved from the NIH NCBI database [22].

**Figure 4 ijms-26-00279-f004:**
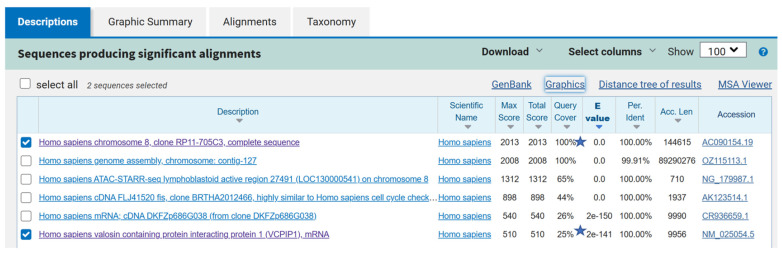
Validation of the *VCPIP1* promoter and 5’ untranslated sequence using the BLAST^TM^ program. The sequences corresponding to the promoter and the 5’ untranslated region were analyzed using the BLAST^TM^ program (checks and stars; [22]). There is 100% alignment of the query sequence with the chromosome 8 region and 25% alignment with the *VCPIP1* sequence; only a short region *VCPIP1* mRNA sequence is analyzed.

**Figure 5 ijms-26-00279-f005:**
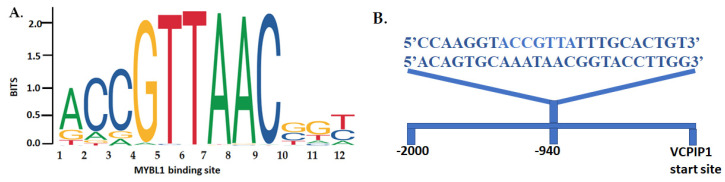
*MYBL1* transcription factor binding sequence. (**A**) The *MYBL1* transcription factor binding sequence was retrieved from the JASPAR online resource [28]. (**B**) The MYBL1 transcription factor binding sequence in the *VCPIP1* promoter region is ~940 base pairs upstream of the protein start site.

**Figure 6 ijms-26-00279-f006:**
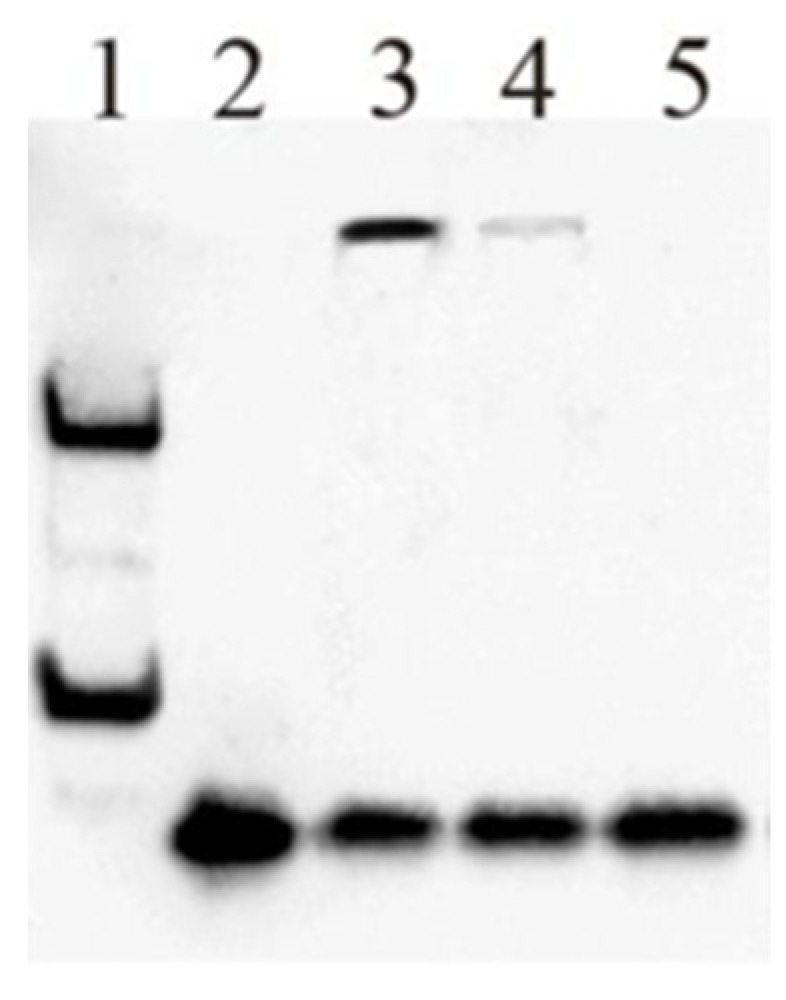
Electrophoretic Mobility Shift Assay analyses demonstrating the ability of MYBL1 protein binding to sequence in the *VCPIP1* promoter region. The EMSA was performed by the Creative-Proteomics company. The blot was exposed for 1 min. Lane 1: Positive Epstein–Barr virus protein probe + Ebstein probe with Positive Ebstein protein (assay control only). Lane 2: Probe 3-Biotin (no MYBL1 protein–promoter binding sequence only). Lane 3: Probe 3-Biotin + MYBL1. Lane 4: Probe 3-Biotin + Probe 3-Cold (50×) + MYBL1 (unlabeled). Lane 5: Probe 3-Biotin + Probe 3-Cold (100×) + MYBL1 (more unlabeled).

**Figure 7 ijms-26-00279-f007:**
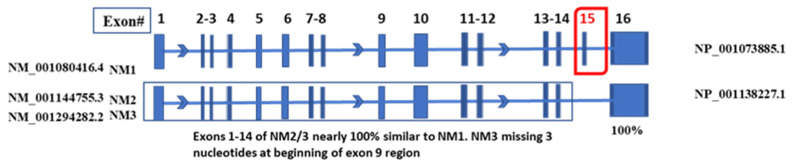
Comparative analyses of the individual exons of the known, curated MYBL1 Reference Sequence transcript variants. Boxes represent exons and lines represent introns for the curated Reference Sequences. The percent similarity of each exon is based on a comparison to the NM1 exon sequences. NM1 is the canonical sequence which contains a unique exon 15. NM2 and NM3 differ by three nucleotides at nucleotide position ~1249 (in NM3). Other than the exon 15 sequence in NM1 and the three nucleotides missing in NM3, all other sequences in MYBL1 are perfectly aligned as determined by MultAlin program [32].

**Figure 8 ijms-26-00279-f008:**
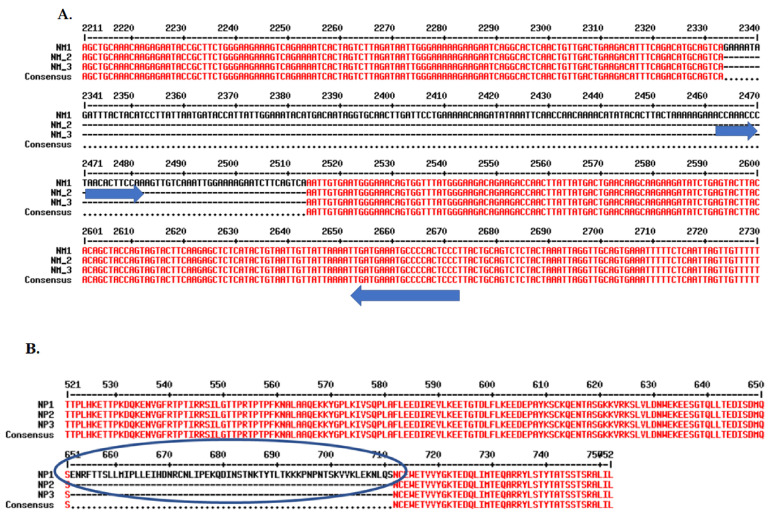
MultAlin sequence alignment comparison between MYBL1 NM1, NM2, and NM3 transcript variants and corresponding isoforms. (**A**) The unique exon 15 region of the transcript is from nucleotide number 2333 to 2513. Exon 15 specific PCR primers are designated by the blue arrows. (**B**) Sequence alignment comparison between the MYBL1 NP1, NP2, and NP3 protein isoforms to demonstrate the location of the amino acids corresponding to the exon 15 region. The amino acids in the oval-shaped area correspond to the exon 15 region. The sequence alignment was performed utilizing the MultAlin program [32]. The numbers above the nucleotide (i.e., NM) and amino acid (i.e., NP) sequences represent the distance of the sequences from their corresponding start sites.

**Figure 9 ijms-26-00279-f009:**
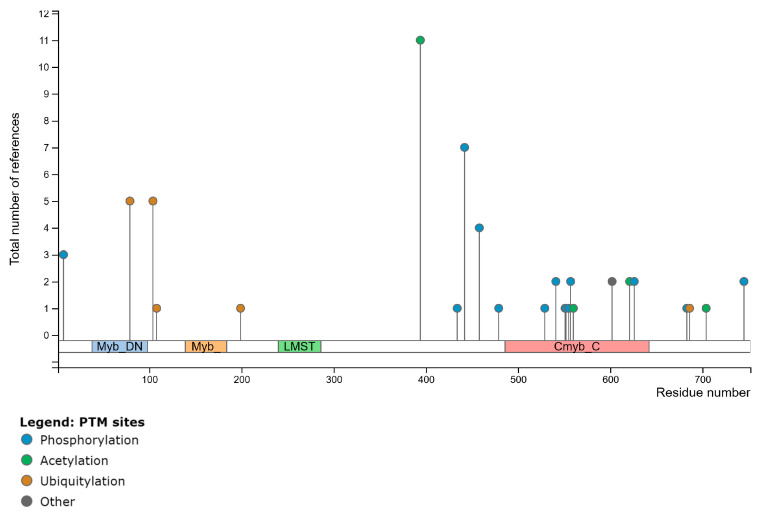
PhosphoSite-PLus^TM^ analyses of the MYBL1 domains and regions susceptible to post-translational modification. Region 650–700 corresponds to amino acids corresponding to the exon 15 region [31].

**Figure 10 ijms-26-00279-f010:**
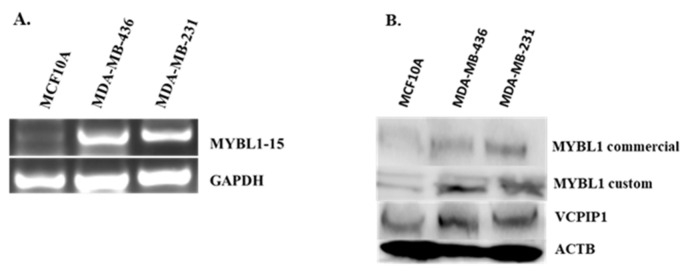
PCR and protein expression analyses of sequences corresponding to MYBL1 exon 15 in control compared to TNBC cells. (**A**) Gene-specific PCR primers were utilized to detect transcript variants expression the MYBL1 exon 15 in non-tumor (MCF10) compared to TNBC (MDA MB436 and MDA MB231) cells. (**B**) Two MYBL1 antibodies were used to detect isoforms expressing the exon 15-region in control compared to TNBC breast cells; a commercial and a custom-made MYBL1 antibody. The commercial MYBL1 antibody recognizes regions corresponding to the exon 15 target, including additional surrounding amino acids. The custom antibody was generated using amino acids unique to the exon 15 region only. Protein expression analyses of the *VCPIP1* gene were included in this experiment.

**Figure 11 ijms-26-00279-f011:**
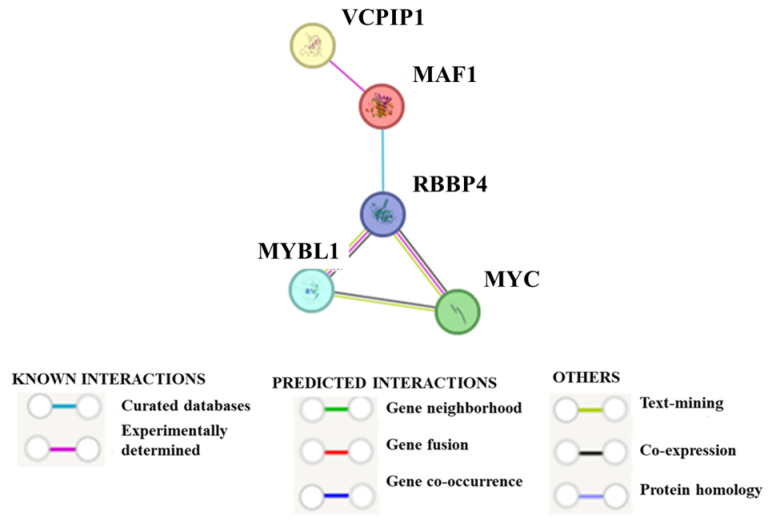
STRING^TM^ analyses showing a possible connection between MYBL1 and VCPIP1. The PPI enrichment *p*-value is 0.00123. Based on STRING, the relationship between MYBL1 and RBBP4 is via co-expression, text-mining and experimentally determined. Based on previous STRING^TM^ analyses, The MAF1 and *VCPIP1* gene interactions were experimentally determined [35].

**Table 1 ijms-26-00279-t001:** The PCR primer sequences utilized to detect *GAPDH*, *MYBL1* and *VCPIP1* transcripts.

GENE	SEQUENCE (LEFT)	SEQUENCE (RIGHT)	AMPLICON SIZE
*GAPDH*	TCCCTGAGCTGAACGGGAAG (L)	GGAGGAGTGGGTGTCGCTGT (R)	217bp
*MYBL1 ORIGINAL*	TGGATAAGTCTGGGCTTATTGG (L)	CCATGCAAGTATGGCTGCTA (R)	210 bp
*MYBL1 (EXON 15)*	ACCAAACCCTAACACTTCCAA (L)	AGGGAGTGGGGCATTTCATC (R)	212 bp
*VCPIP1*	CAGGCAGCTTGATCCTGATT (L)	CTCCCAGTGCATCTGCTACA (R)	272bp

## Data Availability

The data is available from the P.I. (A.P.) upon request.

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
