# Peer review of "Analyses of the MYBL1 Gene in Triple Negative Breast Cancer: Evidence of Regulation of the VCPIP1 Gene and Identification of a Specific Exon Overexpressed in Tumor Cell Lines"

_ijms, 2024, doi:10.3390/ijms26010279_

Round 1

Reviewer 1 Report

Comments and Suggestions for Authors

The aim of the study conducted by Chidinma Nganya et al. is to analyze the MYBL1 gene in triple-negative breast cancer using cell line studies. I believe that the authors have undertaken to clarify an important topic in the context of the genetic characteristics of TNBC. In my opinion, the studies were planned and conducted correctly. The type of experiments was appropriately selected for the undertaken research topic. The results were presented in a reliable and transparent manner. Literature sources were correctly cited. The discussion was conducted in an exhaustive manner, the obtained results were conscientiously compared with already published data.

Please improve the readability of figures: 2a, 11.

In summary, if the authors follow the reviewer's suggestions and improve the quality of the figures presented in the work, I recommend the submitted manuscript for publication in IJMS.

Author Response

Reviewer 1:

Comments and Suggestions for Authors

The aim of the study conducted by Chidinma Nganya et al. is to analyze the MYBL1 gene in triple-negative breast cancer using cell line studies. I believe that the authors have undertaken to clarify an important topic in the context of the genetic characteristics of TNBC. In my opinion, the studies were planned and conducted correctly. The type of experiments was appropriately selected for the undertaken research topic. The results were presented in a reliable and transparent manner. Literature sources were correctly cited. The discussion was conducted in an exhaustive manner, the obtained results were conscientiously compared with already published data.

Please improve the readability of figures: 2a, 11.

Authors response to reviewer: The authors have improved the quality of Figures 2a and Figure 11.

In summary, if the authors follow the reviewer's suggestions and improve the quality of the figures presented in the work, I recommend the submitted manuscript for publication in IJMS.

Reviewer 2 Report

Comments and Suggestions for Authors

The manuscript presents a comprehensive investigation into the role of the MYBL1 gene in triple-negative breast cancer (TNBC), particularly its regulation of the VCPIP1 gene and the identification of a unique exon associated with MYBL1 overexpression in tumor cell lines. While the research provides novel insights into the regulatory mechanisms of MYBL1 and its potential as a biomarker, several areas require clarification, additional references, and improvements in the structure and language.

Comments

Abstract

Line 9: The phrase "we propose that co-expression of the two genes is attributed to MYBL1 transcription factor regulation of the VCPIP1 gene" can be streamlined. Consider rephrasing to "we propose that MYBL1 regulates VCPIP1 expression via transcription factor binding."

Line 15: Expand on the implications of MYBL1's regulation of VCPIP1 in TNBC progression to enhance the abstract's impact.

Introduction

Line 22: Line 20: Provide updated statistics on cancer as well as this cancer type prevalence, including survival rates, to highlight the critical need for prognostic biomarkers. Cite “Cancer statistics, 2024, 2024”. Then give intro in cancer therapy in general, cite NIH paper “Cancer treatments: Past, present, and future, 2024” for more information to contextualize the importance of MYBL1 research. Refer to recent epidemiological studies for this purpose.

Line 40: The role of MYBL1 in cell cycle regulation is briefly mentioned. Include references discussing MYBL1's involvement in the DREAM complex and its downstream targets to add depth.

Results

Line 81: The PCR analysis results comparing MYBL1 and VCPIP1 expression in non-tumor versus TNBC cell lines are presented but lack statistical validation. Include p-values and replicate numbers to support the findings.

Line 130: The identification of the MYBL1 binding site in the VCPIP1 promoter region is intriguing but requires more context. Explain how these binding sites were experimentally validated.

Line 200: Provide further details on the functional implications of exon 15 overexpression in TNBC cell lines. Discuss whether this unique exon contributes to altered protein function or cellular behavior.

Figures

Ensure all figures are referenced in the text. For example, Figures 3 and 4 are not explicitly mentioned in the relevant results section, which diminishes their contextual relevance.

The legends for Figures 7 and 10 should clearly describe the experimental conditions and key findings to aid reader comprehension.

Discussion

Line 251: The relationship between MYBL1 and VCPIP1 in TNBC is discussed but lacks integration with broader literature. Compare these findings with similar regulatory mechanisms in other cancers.

Line 300: Discuss the clinical relevance of MYBL1 as a potential therapeutic target in TNBC, including challenges in targeting transcription factors and unique exon regions. Discuss How these The relationship between MYBL1 and VCPIP1 in TNBC affect immune cells, mention recent immune cell single cell data analysis, and relate these findings to your study, such study includes “Identification of the novel exhausted T cell CD8 + markers in breast cancer, 2024,Genetic Analysis Uncovers Potential Mechanisms Linking Juvenile ldiopathic Arthritisto Breast Cancer: A BioinformaticPilot Study, 2024”

Line 321: Expand on the evolutionary significance of exon variants, referencing studies on transcript diversity and its implications for tumor progression. Suggest future studies that could validate these findings in patient-derived xenograft models or larger cohorts. Previous studies using xenograft models of breast cancer should be mentioned, such as “Comparing volatile and intravenous anesthetics in a mouse model of breast cancer metastasis, 2018”

Materials and Methods

Line 350: Provide more details about the patient sample datasets analyzed, including inclusion criteria and data normalization steps.

Line 420: Describe how the custom antibody for MYBL1 exon 15 was validated to ensure specificity.

Author Response

Dear reviewer---Authors’ responses are included below, but in addition we have included as separate but combined docs (a) evidence of the PCR statistics and (c) examples of the PCR validation to show examples of profiles comparing VCPIP1 vs MYBL1 transcript levels.

Reviewer 2

Comments and Suggestions for Authors

The manuscript presents a comprehensive investigation into the role of the MYBL1 gene in triple-negative breast cancer (TNBC), particularly its regulation of the VCPIP1 gene and the identification of a unique exon associated with MYBL1 overexpression in tumor cell lines. While the research provides novel insights into the regulatory mechanisms of MYBL1 and its potential as a biomarker, several areas require clarification, additional references, and improvements in the structure and language.

Comments

Abstract

Line 9: The phrase "we propose that co-expression of the two genes is attributed to MYBL1 transcription factor regulation of the VCPIP1 gene" can be streamlined. Consider rephrasing to "we propose that MYBL1 regulates VCPIP1 expression via transcription factor binding."

Authors response- We appreciate the Reviewer’s suggestion, but we feel strongly about this statement. A number of analyses have been performed as validation of the relationship between MYBL1 and VCPIP1. The various different analyses (i.e., targeted knockdown study, PCR, and analyses of clinical patient datasets) show co-expression of MYBL1 and VCPIP1 genes. Because MYBL1 is a transcription factor and functions to regulate the expression of other genes, ‘we propose that co-expression is indeed attributed to the ability of MYBL1 to transcriptionally regulate VCPIP1’. We would sincerely like to keep this statement as its written because it more accurately states our thoughts on the process.

Line 15: Expand on the implications of MYBL1's regulation of VCPIP1 in TNBC progression to enhance the abstract's impact.

Authors included this statement- VCPIP1 gene functions as a deubiquitinating enzyme involved in DNA repair, protein positioning, and the assembly of the Golgi apparatus during mitotic signaling. Transcriptional regulation of VCPIP1 by MYBL1 gene could implicate MYBL1 in these processes which might contribute to tumor processes in TNBC.

Introduction

Line 22: Line 20: Provide updated statistics on cancer as well as this cancer type prevalence, including survival rates, to highlight the critical need for prognostic biomarkers. Cite “Cancer statistics, 2024, 2024”. Then give intro in cancer therapy in general, cite NIH paper “Cancer treatments: Past, present, and future, 2024” for more information to contextualize the importance of MYBL1 research. Refer to recent epidemiological studies for this purpose.

Authors included this statement: In the United States of America alone, in the year 2024, its estimated that 2,001,140 new patients will be diagnosed with cancer. Of these cases, 310,720 will be invasive breast cancers, with an estimated 10-15% of these patients diagnosed with TNBC [1]. While effective targeted therapies exist for receptor-positive breast cancers, fewer targeted therapeutic options exist for patients with TNBC.  More recently, immuno-therapy has been approved and show promise in the treatment of some TNBCs [2]. Even with an effective therapy it is still critically important to understand the underlying molecular events that drive TNBC processes so that additional biomarkers can be identified.

Line 40: The role of MYBL1 in cell cycle regulation is briefly mentioned. Include references discussing MYBL1's involvement in the DREAM complex and its downstream targets to add depth.

Authors included this statement (italic already in text; other sentences new):  Specifically related to MYBL1, it is one of several genes associated with the DREAM complex which functions to repress genes involved in cell cycle regulation [6,7]. DREAM stands for ‘dimerization partner, RB-like, E2F, and multi-vulva class B (MuvB)’. The complex includes the E2F4, E2F5, LIN9, LIN37, LIN52, LIN54, MYBL1, MYBL2, RBL1, RBL2, RBBP4, TFDP1 and TFDP2 genes of which MYBL1 is a member. Collectively the genes are master regulators of the cell cycle at stages G1/S and G2/M where the RBL1 and RBL2 are key regulators in the protein complex. During repressive and activating phases of the cell cycle various members of the complex function by binding or dissociating with each other, leading to regulation of the cell cycle. Recent data show that both MYBL1 and MYBL2 can bind MuvB (which includes the LIN9, LIN37, LIN52, LIN54, and RBBP4 complex), leading to downstream recruitment of the other cell cycle regulators CDK1, CCNB1 and FOXM1 genes [8]. These data demonstrate the importance of MYBL1 in a cell cycle regulation, a key process related to the pathogenesis of tumors.

Results

Line 81: The PCR analysis results comparing MYBL1 and VCPIP1 expression in non-tumor versus TNBC cell lines are presented but lack statistical validation. Include p-values and replicate numbers to support the findings.

Authors included this statement: In 11 independent PCR experiments, the difference between the mean densitometer values for VCPIP1 expression in MCF10A control versus TNBC was 0.28 compared to 0.78, respectively, with a p-value of 0.02. And the difference between the mean densitometer values for MYBL1 expression in MCF10A control versus TNBC was 0.25 versus 0.61, respectively, with a p-value of 0.03 (data not shown).

In addition the authors have included (as an attachment to the ‘Response document’, the raw values used to generate the Statistics referenced above and examples of some of the actual PCRs. Excuse the volume of information, but we found it necessary to include these docs to the Reviewer.

Line 130: The identification of the MYBL1 binding site in the VCPIP1 promoter region is intriguing but requires more context. Explain how these binding sites were experimentally validated.

Authors included this statement: Before a transcription factor can regulate expression of a particular gene, the protein must first bind to the promoter region of the targeted gene. The EMSA assay is performed to show that the MYBL1 transcription factor can bind to the promoter region of the VCPIP1 gene. The purified MYBL1 protein only bound to 1 of the putative VCPIP1 sequences (Figure 6); the sequence given in Figure 5b. To validate the performance of the EMSA assay, as control the Epstein-Barr virus protein was shown to bind to its specific DNA probe sequence (lane 1). Lane 2 contains the biotin end-labeled VCPIP1 probe given in Figure 5b. Without MYBL1, the biotin-labeled VCPIP1 probe migrates as a low molecular weight sequence (lane 2) but following incubation with and binding of the unlabeled MYBL1 recombinant protein, the mobility of the VCPIP1 fragment shows a shift in mobility and higher molecular weight migration of the probe. These data show that the MYBL1 protein can bind to the VCPIP1 promoter region.  An increase in the concentration of a ‘cold unlabeled VCPIP1 probe binding to the unlabeled MYBL1 protein’ was utilized as an EMSA control, to demonstrate a lack in the ‘shift in the mobility of the VCPIP1 probe’ when high concentrations of the unlabeled probe competes with binding MYBL1 protein. Experimental results of the EMSA assay show that the purified MYBL1 recombinant protein binds to the sequence ~940 nucleotides upstream of the VCPIP1 start site. These results support MYBL1 protein’s ability to bind to the promoter region of the VCPIP1 gene.

Line 200: Provide further details on the functional implications of exon 15 overexpression in TNBC cell lines. Discuss whether this unique exon contributes to altered protein function or cellular behavior.

Authors included this statement: Data show that individual isoforms for a particular gene can have different functions in the same type of sample emphasizing the importance of characterizing the isoforms related to particular genes [30]. We screened several commercially available MYBL1 antibodies and found that antibodies that recognized the region corresponding to the exon 15, demonstrated consistent differential detection of TNBC samples. At this stage of the project, we do not know how exon-15 contributes to the tumor process, only that exon 15 is over represented in the TNBC compared to the non-tumor breast cells. For any particular gene, generally the transcript variants are functionally related. The variants might have opposing functions, but generally there is some relationship. For MYBL1 gene we suspect that the function of variants without or with exon 15 differ based on transcriptional activation because MYBL1 gene is a transcription factor.   

Figures

Ensure all figures are referenced in the text. For example, Figures 3 and 4 are not explicitly mentioned in the relevant results section, which diminishes their contextual relevance.

Authors response and included the statement below—All figures are referenced in the text. But we agree with the Reviewer in that all of the figures are not explicitly described.  

Added this statement: Using resources available on the NCBI website, we identified the VCPIP1 promoter region upstream of the gene’s start site on chromosome 8 (Figure 3).  The start site for the VCPIP1 protein is designated by the ATG nucleotide sequence. In the event MYBL1 regulates VCPIP1, the MYBL1 protein binding site should be present in the VCPIP1 promoter region. The VCPIP1 promoter region was retrieved from the NCBI website and validated by sequence alignment analyses. The BLAST program was utilized to validate the VCPIP1 promoter sequence given in Figure 3.  The sequence given in Figure 3 aligns 100% with chromosome 8 locus corresponding to the VCPIP1 promoter, and 25% with the actual VCPIP1 gene (Figure 4). The entire VCPIP1 gene was not analyzed via BLAST, only 25% of the VCPIP1 gene was analyzed using BLAST.

The legends for Figures 7 and 10 should clearly describe the experimental conditions and key findings to aid reader comprehension.

Authors clarified the description of Figure 7 and Figure 10: Included these statements:

Figure 7: Comparative analyses of the individual exons of the known-curated MYBL1 Reference Sequence transcript variants. Boxes represent exons and lines represent introns for the curated Reference Sequences. The percent similarity of each exon is based on comparison to the NM1 exon sequences. NM1 is the canonical sequence which contains a unique exon 15. NM2 and NM3 differ by 3 nucleotides at nucleotide position ~1249 (in NM3). Other than the exon 15 sequence in NM1 and 3 nucleotides missing in NM3, all other sequences in MYBL1 are perfectly aligned [16].

Figure 10: PCR and protein expression analyses of sequences corresponding to MYBL1 exon 15 in control compared to TNBC cells. (a) gene specific PCR primers were utilized to detect transcript variants expression the MYBL1 exon 15 in non-tumor (MCF10) compared to TNBC (MDA MB436 and MDA MB231) cells. (b) two MYBL1 antibodies were used to detect isoforms expressing the exon 15 region in control compared to TNBC breast cells; a commercial and a custom-made MYBL1 antibody. The commercial MYBL1 antibody recognizes regions corresponding to the exon 15 target including additional surrounding amino acids. The custom antibody was generated using amino acids unique to the exon 15 region only. Protein expression analyses of VCPIP1 gene was included in this experiment.

Discussion

Line 251: The relationship between MYBL1 and VCPIP1 in TNBC is discussed but lacks integration with broader literature. Compare these findings with similar regulatory mechanisms in other cancers.

Author’s response to Reviewer--- Please see lines 28-32 (before revision); authors also added an additional passage (included below) to reference a relationship between MYBL1 and immune system.

‘The MYBL1 gene is over-expressed in low grade gliomas [3], dysregulated in breast adenoid cystic carcinomas (a rare triple negative breast cancer (TNBC)) and salivary gland carcinomas where gene-fusion mutations with ACTN1 and NFIB genes are identified [6,7]. MYBL1 gene is also over-expressed in clear cell renal carcinoma and considered an immunotherapeutic biomarker for these cancers [8].’

In addition to address the relationship to immune-markers we have included this following statement at the end of this passage above:

MYBL1 gene is also over-expressed in clear cell renal carcinoma and considered an immunotherapeutic biomarker for these cancers [8].’ Studies of clear cell renal carcinoma show that MYBL1 expression correlations with the immune scores, increasing Tregs, M2 macrophages, neutrophils, B cells, monocytes and CD8+ Tcells [12]. Over-expression of MYBL1 also correlates with overexpression of the key immune checkpoint genes PD-1, CTLA4, PD-L1 and PD-L2 in clear cell renal cell carcinoma patients. The authors suggest that MYBL1 can ultimately remodel the tumor micro-environment, but it’s unclear how MYBL1 might enhance the cellular malignant behaviors of clear cell renal carcinomas.

Line 300: Discuss the clinical relevance of MYBL1 as a potential therapeutic target in TNBC, including challenges in targeting transcription factors and unique exon regions. Discuss How these The relationship between MYBL1 and VCPIP1 in TNBC affect immune cells, mention recent immune cell single cell data analysis, and relate these findings to your study, such study includes “Identification of the novel exhausted T cell CD8 + markers in breast cancer, 2024,Genetic Analysis Uncovers Potential Mechanisms Linking Juvenile ldiopathic Arthritisto Breast Cancer: A BioinformaticPilot Study, 2024”

Author responses to reviewer—Dear Reviewer, we do not think that nuclear proteins are suitable for targeting, and do not propose that MYBL1 be considered for targeting. As you are probably aware, cell surface or diffusible factors are more suited. We do not think that MYBL1 nuclear protein is a suitable biomarker for therapeutic targeting. If MYBL1 were dysregulated in a larger number of TNBC, we would perform analyses to determine clinical significance of over-expression of the gene, but MYBL1 (as mentioned in our original text) is dysregulated in a subcategory (not all) of the TNBC, and TNBC represent ~15% of breast cancers. Please excuse us, but our manuscript is a research manuscript and at this phase of the research we do not see MYBL1 as a target with clinical implications. Based on our studies we see MYBL1 as a gene possibly expressed early in tumor progression; a gene that regulates late stage genes like possible MYC and maybe even TP53. From (a) our studies, (b) our approaches and (c) the fact that MYBL1 is known to regulate and be involved in events key to tumors, we absolutely know that we can identify biomarkers with some utility as a biomarker, OR targeting , but we are not there yet.

Also, we mentioned in the ABSTRACT-line 15: Although both genes are involved in cell cycle regulatory mechanisms, converging signaling mechanisms have not been identified.

And on Line 278 (before revisions): Investigators show that VCPIP1 interacts with VCP and other genes associated with reassembly of the Golgi apparatus and the endoplasmic reticulum during mitosis [15,31]. We examined the possibility that MYBL1 might be associated with this sig-naling pathway, but the experimental results were inconsistent.-

Data not shown: We chose to examine this Golgi signaling pathway because associated genes were identified as affected by the knockdown of MYBL1 in MDA MB231 cells. Even though the results were inconsistent, we are not convinced that MYBL1 and VCPIP1 cooperate in these signaling mechanisms.

To Reviewer the following passage was included: As we appreciate the Reviewers’ suggestion to address the immune-relationship… but please excuse us, but we think it more appropriate to address a relationship between immunity and MYBL1. Also noted above:

MYBL1 gene is also over-expressed in clear cell renal carcinoma and considered an immunotherapeutic biomarker for these cancers [8].’ Studies of clear cell renal carcinoma show that MYBL1 expression correlations with the immune scores, increasing Tregs, M2 macrophages, neutrophils, B cells, monocytes and CD8+ Tcells [12]. Over-expression of MYBL1 also correlates with overexpression of the key immune checkpoint genes PD-1, CTLA4, PD-L1 and PD-L2 in clear cell renal cell carcinoma patients. The authors suggest that MYBL1 can ultimately remodel the tumor micro-environment, but it’s unclear how MYBL1 might enhance the cellular malignant behaviors of clear cell renal carcinomas.                      

Line 321: Expand on the evolutionary significance of exon variants, referencing studies on transcript diversity and its implications for tumor progression. Suggest future studies that could validate these findings in patient-derived xenograft models or larger cohorts. Previous studies using xenograft models of breast cancer should be mentioned, such as “Comparing volatile and intravenous anesthetics in a mouse model of breast cancer metastasis, 2018”

Authors included this statement—‘Different isoforms of the same gene are thought to be an evolutionary strategy to promote molecular diversity and/or exist where one isoform, via competitive binding, can modulate the function of the other in vivo [38]. Wang and Kirkness [44] suggest that in mammals, variations in the number of exons lead to lineage-specific evolution. Nearly all genes exist as transcript variations and subsequent protein isoforms, and data show that the variations of the particular gene do not always have the same function. Gene variations are generally related to splicing events that lead to differences in the exon pattern of the gene, leading to disease processes, including associations with tumors [45,46]. Under these conditions, the exons are described as tumor-associated exons. Researchers understand the slicing mechanisms that lead to variations related to exon differences, but not the underlying events that trigger the processes. Once the triggering event has occurred, in the case of cancers, it could be that at the early stage of tumor progression, there were higher levels of the non-tumor variants compared to variants expressing tumor-associated exons. And with later stages of progression, there is a shift such that higher levels of the variants expressing the tumor associated exons. Regardless, the function of the tumor-associated exon clearly relies upon the function of the gene.

In addition: The reviewer suggests that xenografts be performed. Before these experiments are done we feel that it more important to first expand our probe of TNBC cell lines for expression of the exon-15 and corresponding isoform. These experiments are included in our original manuscript. See lines 340-344. After these experiments are completed, we can then consider introduction of the exon-15 expressing into non-tumor cells can be assessed as xenografts experiments.

Materials and Methods

Line 350: Provide more details about the patient sample datasets analyzed, including inclusion criteria and data normalization steps.

Author response to the reviewer:  Curated datasets were available via the cBioPortal.org Cancer Genomics website, and analyzed using the cBioPortal.org online data analysis resources. As I am sure that the reviewer is aware, online datasets (ie, NBCI, TCGA, cBioPortal, etc…) are nearly always normalized prior to depositing. So, the researcher is free to perform meta-analyses. We mention in the Methods that the datasets were filtered based on receptor-status, and only TNBC samples were analyzed for alterations in MYBL1 and VCPIP1 genes.

Line 420: Describe how the custom antibody for MYBL1 exon 15 was validated to ensure specificity.

Author response to the reviewer- the following passage was included: The custom antibody for MYBL1 gene was based on the amino acid sequence corresponding to exon 15 only. The following amino acid sequence corresponds to the exon 15 nucleotide sequence: ENRFTTSLLMIPLLEIHDNRCNLIPEKQDINSTNKTYTLT-KKKPNPNTSKVVKLEKNLQS.  Note that this immunogen is the same as the amino acids underlined for the commercial MYBL1 antibody. The sequence was made available to Sino Biotechnologies company. The company generated the amino acid sequence, injected it into rabbits, bleed the rabbits after 6 months, purified the IgG antibody and shipped the antibody protein to our university. Based on the immunogen, only variants with amino acids corresponding to the exon 15 region are detected. And only protein isoforms containing the immunogen sequence should bind the antibody. Full length MYBL1 with exon 15 is ~85,885 Daltons. The western blot was performed as validation and we observed an 85,000 Dalton protein band corresponding to  MYBL1 isoform for the protein containing amino acids corresponding to exon 15 region. The molecular weight for the reactive protein was determined by comparison to Precision Plus Protein™ Dual Color Standard molecular weight standard (catalog # 1610374 ; Bio-Rad, Hercules, CA, USA).

Round 2

Reviewer 2 Report

Comments and Suggestions for Authors

The paper is been improved. Part of the issue is been addressed, and the author do not willing to address others but gives sufficient explanation. I believe these issue still should be discussed as least in the discussion, for example, suggestion of future animal studies, and in the discussion, they can put whatever they explain to me. How these can be further improved. In addition, I still strongly suggest to cite NIH paper “Cancer treatments: Past, present, and future, 2024” (or similar alternative)) for more information to provide general background in cancer therapy for general readers. Reason is that you need reader to understand why you are still doing cancer research, what you aim to overcome in this study.

Author Response

Dear reviewer---Authors’ responses are included below, but in addition we have included as separate but combined docs (a) evidence of the PCR statistics and (c) examples of the PCR validation to show examples of profiles comparing VCPIP1 vs MYBL1 transcript levels.

Also, thank you for your comments. Whether or not you choose to publish our work, addition of your comments substantially improves our manuscript. Sincerely, thank you.

Reviewer 2

Comments and Suggestions for Authors

The manuscript presents a comprehensive investigation into the role of the MYBL1 gene in triple-negative breast cancer (TNBC), particularly its regulation of the VCPIP1 gene and the identification of a unique exon associated with MYBL1 overexpression in tumor cell lines. While the research provides novel insights into the regulatory mechanisms of MYBL1 and its potential as a biomarker, several areas require clarification, additional references, and improvements in the structure and language.

Comments

Abstract

Line 9: The phrase "we propose that co-expression of the two genes is attributed to MYBL1 transcription factor regulation of the VCPIP1 gene" can be streamlined. Consider rephrasing to "we propose that MYBL1 regulates VCPIP1 expression via transcription factor binding."

Authors response- Line 11. We appreciate the Reviewer’s suggestion, but we feel strongly about this statement. A number of analyses have been performed as validation of the relationship between MYBL1 and VCPIP1. The various different analyses (i.e., targeted knockdown study, PCR, and analyses of clinical patient datasets) show co-expression of MYBL1 and VCPIP1 genes. Because MYBL1 is a transcription factor and functions to regulate the expression of other genes, ‘we propose that co-expression is indeed attributed to the ability of MYBL1 to transcriptionally regulate VCPIP1’. We would sincerely like to keep this statement as its written because it more accurately states our thoughts on the process.

Line 15: Expand on the implications of MYBL1's regulation of VCPIP1 in TNBC progression to enhance the abstract's impact.

Authors included this statement- Line 15. VCPIP1 gene functions as a deubiquitinating enzyme involved in DNA repair, protein positioning, and the assembly of the Golgi apparatus during mitotic signaling. Transcriptional regulation of VCPIP1 by MYBL1 gene could implicate MYBL1 in these processes which might contribute to tumor processes in TNBC.

Introduction

Line 22: Line 20: Provide updated statistics on cancer as well as this cancer type prevalence, including survival rates, to highlight the critical need for prognostic biomarkers. Cite “Cancer statistics, 2024, 2024”. Then give intro in cancer therapy in general, cite NIH paper “Cancer treatments: Past, present, and future, 2024” for more information to contextualize the importance of MYBL1 research. Refer to recent epidemiological studies for this purpose.

Authors included this statement: line 25. In the United States of America alone, in the year 2024, its estimated that 2,001,140 new patients will be diagnosed with cancer. Of these cases, 310,720 will be invasive breast cancers, with an estimated 10-15% of these patients diagnosed with TNBC [1]. While effective targeted therapies exist for receptor-positive breast cancers, fewer targeted therapeutic options exist for patients with TNBC.  More recently, immuno-therapy has been approved and show promise in the treatment of some TNBCs [2]. Even with an effective therapy it is still critically important to understand the underlying molecular events that drive TNBC processes so that additional biomarkers can be identified.

Line 40: The role of MYBL1 in cell cycle regulation is briefly mentioned. Include references discussing MYBL1's involvement in the DREAM complex and its downstream targets to add depth.

Authors included this statement (italic already in text; other sentences new):  Specifically related to MYBL1, it is one of several genes associated with the DREAM complex which functions to repress genes involved in cell cycle regulation [6,7]. Line 39. DREAM stands for ‘dimerization partner, RB-like, E2F, and multi-vulva class B (MuvB)’. The complex includes the E2F4, E2F5, LIN9, LIN37, LIN52, LIN54, MYBL1, MYBL2, RBL1, RBL2, RBBP4, TFDP1 and TFDP2 genes of which MYBL1 is a member. Collectively the genes are master regulators of the cell cycle at stages G1/S and G2/M where the RBL1 and RBL2 are key regulators in the protein complex. During repressive and activating phases of the cell cycle various members of the complex function by binding or dissociating with each other, leading to regulation of the cell cycle. Recent data show that both MYBL1 and MYBL2 can bind MuvB (which includes the LIN9, LIN37, LIN52, LIN54, and RBBP4 complex), leading to downstream recruitment of the other cell cycle regulators CDK1, CCNB1 and FOXM1 genes [8]. These data demonstrate the importance of MYBL1 in a cell cycle regulation, a key process related to the pathogenesis of tumors.

Results

Line 81: The PCR analysis results comparing MYBL1 and VCPIP1 expression in non-tumor versus TNBC cell lines are presented but lack statistical validation. Include p-values and replicate numbers to support the findings.

Authors included this statement: line 107. In 11 independent PCR experiments, the difference between the mean densitometer values for VCPIP1 expression in MCF10A control versus TNBC was 0.28 compared to 0.78, respectively, with a p-value of 0.02. And the difference between the mean densitometer values for MYBL1 expression in MCF10A control versus TNBC was 0.25 versus 0.61, respectively, with a p-value of 0.03 (data not shown).

In addition the authors have included (as an attachment to the ‘Response document’, the raw values used to generate the Statistics referenced above and examples of some of the actual PCRs. Excuse the volume of information, but we found it necessary to include these docs to the Reviewer.

Line 130: The identification of the MYBL1 binding site in the VCPIP1 promoter region is intriguing but requires more context. Explain how these binding sites were experimentally validated.

Authors included this statement: line 187. Before a transcription factor can regulate expression of a particular gene, the protein must first bind to the promoter region of the targeted gene. The EMSA assay is performed to show that the MYBL1 transcription factor can bind to the promoter region of the VCPIP1 gene. The purified MYBL1 protein only bound to 1 of the putative VCPIP1 sequences (Figure 6); the sequence given in Figure 5b. To validate the performance of the EMSA assay, as control the Epstein-Barr virus protein was shown to bind to its specific DNA probe sequence (lane 1). Lane 2 contains the biotin end-labeled VCPIP1 probe given in Figure 5b. Without MYBL1, the biotin-labeled VCPIP1 probe migrates as a low molecular weight sequence (lane 2) but following incubation with and binding of the unlabeled MYBL1 recombinant protein, the mobility of the VCPIP1 fragment shows a shift in mobility and higher molecular weight migration of the probe. These data show that the MYBL1 protein can bind to the VCPIP1 promoter region.  An increase in the concentration of a ‘cold unlabeled VCPIP1 probe binding to the unlabeled MYBL1 protein’ was utilized as an EMSA control, to demonstrate a lack in the ‘shift in the mobility of the VCPIP1 probe’ when high concentrations of the unlabeled probe competes with binding MYBL1 protein. Experimental results of the EMSA assay show that the purified MYBL1 recombinant protein binds to the sequence ~940 nucleotides upstream of the VCPIP1 start site. These results support MYBL1 protein’s ability to bind to the promoter region of the VCPIP1 gene.

Line 200: Provide further details on the functional implications of exon 15 overexpression in TNBC cell lines. Discuss whether this unique exon contributes to altered protein function or cellular behavior.

Authors included this statement: line 260. Data show that individual isoforms for a particular gene can have different functions in the same type of sample emphasizing the importance of characterizing the isoforms related to particular genes [30]. We screened several commercially available MYBL1 antibodies and found that antibodies that recognized the region corresponding to the exon 15, demonstrated consistent differential detection of TNBC samples. At this stage of the project, we do not know how exon-15 contributes to the tumor process, only that exon 15 is over represented in the TNBC compared to the non-tumor breast cells. For any particular gene, generally the transcript variants are functionally related. The variants might have opposing functions, but generally there is some relationship. For MYBL1 gene we suspect that the function of variants without or with exon 15 differ based on transcriptional activation because MYBL1 gene is a transcription factor.   

Figures

Ensure all figures are referenced in the text. For example, Figures 3 and 4 are not explicitly mentioned in the relevant results section, which diminishes their contextual relevance.

Authors response and included the statement below—All figures are referenced in the text. But we agree with the Reviewer in that all of the figures are not explicitly described.  

Added this statement: line 161.Using resources available on the NCBI website, we identified the VCPIP1 promoter region upstream of the gene’s start site on chromosome 8 (Figure 3).  The start site for the VCPIP1 protein is designated by the ATG nucleotide sequence. In the event MYBL1 regulates VCPIP1, the MYBL1 protein binding site should be present in the VCPIP1 promoter region. The VCPIP1 promoter region was retrieved from the NCBI website and validated by sequence alignment analyses. The BLAST program was utilized to validate the VCPIP1 promoter sequence given in Figure 3.  The sequence given in Figure 3 aligns 100% with chromosome 8 locus corresponding to the VCPIP1 promoter, and 25% with the actual VCPIP1 gene (Figure 4). The entire VCPIP1 gene was not analyzed via BLAST, only 25% of the VCPIP1 gene was analyzed using BLAST.

The legends for Figures 7 and 10 should clearly describe the experimental conditions and key findings to aid reader comprehension.

Authors clarified the description of Figure 7 and Figure 10: Included these statements:

Line 242. Figure 7: Comparative analyses of the individual exons of the known-curated MYBL1 Reference Sequence transcript variants. Boxes represent exons and lines represent introns for the curated Reference Sequences. The percent similarity of each exon is based on comparison to the NM1 exon sequences. NM1 is the canonical sequence which contains a unique exon 15. NM2 and NM3 differ by 3 nucleotides at nucleotide position ~1249 (in NM3). Other than the exon 15 sequence in NM1 and 3 nucleotides missing in NM3, all other sequences in MYBL1 are perfectly aligned [16].

Line 291. Figure 10: PCR and protein expression analyses of sequences corresponding to MYBL1 exon 15 in control compared to TNBC cells. (a) gene specific PCR primers were utilized to detect transcript variants expression the MYBL1 exon 15 in non-tumor (MCF10) compared to TNBC (MDA MB436 and MDA MB231) cells. (b) two MYBL1 antibodies were used to detect isoforms expressing the exon 15 region in control compared to TNBC breast cells; a commercial and a custom-made MYBL1 antibody. The commercial MYBL1 antibody recognizes regions corresponding to the exon 15 target including additional surrounding amino acids. The custom antibody was generated using amino acids unique to the exon 15 region only. Protein expression analyses of VCPIP1 gene was included in this experiment.

Discussion

Line 251: The relationship between MYBL1 and VCPIP1 in TNBC is discussed but lacks integration with broader literature. Compare these findings with similar regulatory mechanisms in other cancers.

Author’s response to Reviewer--- Please see lines 50; authors also added an additional passage (included below) to reference a relationship between MYBL1 and immune system.

‘The MYBL1 gene is over-expressed in low grade gliomas [3], dysregulated in breast adenoid cystic carcinomas (a rare triple negative breast cancer (TNBC)) and salivary gland carcinomas where gene-fusion mutations with ACTN1 and NFIB genes are identified [6,7]. MYBL1 gene is also over-expressed in clear cell renal carcinoma and considered an immunotherapeutic biomarker for these cancers [8].’

In addition to address the relationship to immune-markers we have included this following statement at the end of this passage above:

Line 54. MYBL1 gene is also over-expressed in clear cell renal carcinoma and considered an immunotherapeutic biomarker for these cancers [8].’ Studies of clear cell renal carcinoma show that MYBL1 expression correlations with the immune scores, increasing Tregs, M2 macrophages, neutrophils, B cells, monocytes and CD8+ Tcells [12]. Over-expression of MYBL1 also correlates with overexpression of the key immune checkpoint genes PD-1, CTLA4, PD-L1 and PD-L2 in clear cell renal cell carcinoma patients. The authors suggest that MYBL1 can ultimately remodel the tumor micro-environment, but it’s unclear how MYBL1 might enhance the cellular malignant behaviors of clear cell renal carcinomas.

Line 300: Discuss the clinical relevance of MYBL1 as a potential therapeutic target in TNBC, including challenges in targeting transcription factors and unique exon regions. Discuss How these The relationship between MYBL1 and VCPIP1 in TNBC affect immune cells, mention recent immune cell single cell data analysis, and relate these findings to your study, such study includes “Identification of the novel exhausted T cell CD8 + markers in breast cancer, 2024,Genetic Analysis Uncovers Potential Mechanisms Linking Juvenile ldiopathic Arthritisto Breast Cancer: A BioinformaticPilot Study, 2024”

Author responses to reviewer—Dear Reviewer, we do not think that nuclear proteins are suitable for targeting, and do not propose that MYBL1 be considered for targeting. As you are probably aware, cell surface or diffusible factors are more suited. We do not think that MYBL1 nuclear protein is a suitable biomarker for therapeutic targeting. If MYBL1 were dysregulated in a larger number of TNBC, we would perform analyses to determine clinical significance of over-expression of the gene, but MYBL1 (as mentioned in our original text) is dysregulated in a subcategory (not all) of the TNBC, and TNBC represent ~15% of breast cancers. Please excuse us, but our manuscript is a research manuscript and at this phase of the research we do not see MYBL1 as a target with clinical implications. Based on our studies we see MYBL1 as a gene possibly expressed early in tumor progression; a gene that regulates late stage genes like possible MYC and maybe even TP53. From (a) our studies, (b) our approaches and (c) the fact that MYBL1 is known to regulate and be involved in events key to tumors, we absolutely know that we can identify biomarkers with some utility as a biomarker, OR targeting , but we are not there yet.

Also, we mentioned in the ABSTRACT-line 18: Although both genes are involved in cell cycle regulatory mechanisms, converging signaling mechanisms have not been identified.

And on Line 345: Investigators show that VCPIP1 interacts with VCP and other genes associated with reassembly of the Golgi apparatus and the endoplasmic reticulum during mitosis [15,31]. We examined the possibility that MYBL1 might be associated with this sig-naling pathway, but the experimental results were inconsistent.-

Data not shown: We chose to examine this Golgi signaling pathway because associated genes were identified as affected by the knockdown of MYBL1 in MDA MB231 cells. Even though the results were inconsistent, we are not convinced that MYBL1 and VCPIP1 cooperate in these signaling mechanisms.

To Reviewer the following passage was included: As we appreciate the Reviewers’ suggestion to address the immune-relationship… but please excuse us, but we think it more appropriate to address a relationship between immunity and MYBL1. Also noted above:

Line 54. MYBL1 gene is also over-expressed in clear cell renal carcinoma and considered an immunotherapeutic biomarker for these cancers [8].’ Studies of clear cell renal carcinoma show that MYBL1 expression correlations with the immune scores, increasing Tregs, M2 macrophages, neutrophils, B cells, monocytes and CD8+ Tcells [12]. Over-expression of MYBL1 also correlates with overexpression of the key immune checkpoint genes PD-1, CTLA4, PD-L1 and PD-L2 in clear cell renal cell carcinoma patients. The authors suggest that MYBL1 can ultimately remodel the tumor micro-environment, but it’s unclear how MYBL1 might enhance the cellular malignant behaviors of clear cell renal carcinomas.                      

Line 321: Expand on the evolutionary significance of exon variants, referencing studies on transcript diversity and its implications for tumor progression. Suggest future studies that could validate these findings in patient-derived xenograft models or larger cohorts. Previous studies using xenograft models of breast cancer should be mentioned, such as “Comparing volatile and intravenous anesthetics in a mouse model of breast cancer metastasis, 2018”

Authors included this statementLine 394. ‘Different isoforms of the same gene are thought to be an evolutionary strategy to promote molecular diversity and/or exist where one isoform, via competitive binding, can modulate the function of the other in vivo [38]. Wang and Kirkness [44] suggest that in mammals, variations in the number of exons lead to lineage-specific evolution. Nearly all genes exist as transcript variations and subsequent protein isoforms, and data show that the variations of the particular gene do not always have the same function. Gene variations are generally related to splicing events that lead to differences in the exon pattern of the gene, leading to disease processes, including associations with tumors [45,46]. Under these conditions, the exons are described as tumor-associated exons. Researchers understand the slicing mechanisms that lead to variations related to exon differences, but not the underlying events that trigger the processes. Once the triggering event has occurred, in the case of cancers, it could be that at the early stage of tumor progression, there were higher levels of the non-tumor variants compared to variants expressing tumor-associated exons. And with later stages of progression, there is a shift such that higher levels of the variants expressing the tumor associated exons. Regardless, the function of the tumor-associated exon clearly relies upon the function of the gene.

In addition: The reviewer suggests that xenografts be performed. Before these experiments are done we feel that it more important to first expand our probe of TNBC cell lines for expression of the exon-15 and corresponding isoform. These experiments are included in our original manuscript. See lines After these experiments are completed, we can then consider introduction of the exon 15 expressing into non-tumor cells can be assessed as xenografts experiments.

Materials and Methods

Line 350: Provide more details about the patient sample datasets analyzed, including inclusion criteria and data normalization steps.

Author response to the reviewer:  Curated datasets were available via the cBioPortal.org Cancer Genomics website, and analyzed using the cBioPortal.org online data analysis resources. As I am sure that the reviewer is aware, online datasets (ie, NBCI, TCGA, cBioPortal, etc…) are nearly always normalized prior to depositing. So, the researcher is free to perform meta-analyses. We mention in the Methods that the datasets were filtered based on receptor-status, and only TNBC samples were analyzed for alterations in MYBL1 and VCPIP1 genes. Line 440.

Line 420: Describe how the custom antibody for MYBL1 exon 15 was validated to ensure specificity.

Author response to the reviewer- the following passage was included: Line 497.The custom antibody for MYBL1 gene was based on the amino acid sequence corresponding to exon 15 only. The following amino acid sequence corresponds to the exon 15 nucleotide sequence: ENRFTTSLLMIPLLEIHDNRCNLIPEKQDINSTNKTYTLT-KKKPNPNTSKVVKLEKNLQS.  Note that this immunogen is the same as the amino acids underlined for the commercial MYBL1 antibody. The sequence was made available to Sino Biotechnologies company. The company generated the amino acid sequence, injected it into rabbits, bleed the rabbits after 6 months, purified the IgG antibody and shipped the antibody protein to our university. Based on the immunogen, only variants with amino acids corresponding to the exon 15 region are detected. And only protein isoforms containing the immunogen sequence should bind the antibody. Full length MYBL1 with exon 15 is ~85,885 Daltons. The western blot was performed as validation and we observed an 85,000 Dalton protein band corresponding to  MYBL1 isoform for the protein containing amino acids corresponding to exon 15 region. The molecular weight for the reactive protein was determined by comparison to Precision Plus Protein™ Dual Color Standard molecular weight standard (catalog # 1610374 ; Bio-Rad, Hercules, CA, USA).

Please see attachment for evidence of our MYBL1/CPIP1 PCR expression
